# Performance Evaluation of Solid-State Laser Gain Module by Measurement of Thermal Effect and Energy Storage

**Daewoong Park** [1] , **Jihoon Jeong** [2] , **Seungjin Hwang** [3] , **Sungyoon Lee** [1,3], **Seryeyohan Cho** [4,5,*]
**and Tae Jun Yu** [1,3,5,*]

1   Department of Advanced Convergence, Handong Global University, Pohang 37554, Korea;
    daewoongpark@handong.edu (D.P.); sungyoon.lee@handong.edu (S.L.)
2   SEMES Co., Ltd., Cheonan 31040, Korea; jihoonjeong04@gmail.com
3   HIL Lab. Inc., Pohang 37563, Korea; seungjin.hwang@hillab.co.kr
4   Department of Information and Communication Engineering, Handong Global University,
    Pohang 37554, Korea
5   Global Green Research and Development Center, Global Institute of Laser Technology,
    Handong Global University, Pohang 37554, Korea
*   Correspondence: sycho@handong.edu (S.C.); taejunyu@handong.edu (T.J.Y.)

**Abstract:** The optimization of solid-state laser cavities requires a deep understanding of the gain module, the most critical laser component. This study proposes a procedure for evaluating the performance of the solid-state laser gain module. The thermal effect and energy storage characteristics are the performance criteria. A normalized heating parameter was calculated as a quantitative indicator of the performance criteria. We proposed a method to quantify the heat dissipated into the gain medium using the wavefront distortion, thermal deformation theory of the gain medium, and the ray transfer matrix method. The suggested procedure was verified by evaluating the flashlamp type Nd:YAG rod gain module, but it can also even be extended to other solid-state laser gain modules by applying the appropriate thermal deformation theory.

**Keywords:** laser gain module; performance evaluation; normalized heating parameter; gain module efficiency

## 1. Introduction

Nd:YAG-based lasers have been widely applied in industrial and research fields [1–8]. The recent switching trend toward diode pumping methods, advantageous for high efficiency, high power, and high stability, seems to call into question the utility of flashlamps. However, flashlamp-pumped Nd:YAG lasers are also profitable for generating high-power and high-energy beams with a lower price per pump power than with laser diodes, so they have consistently been applied to ultra-high-power laser facilities and industrial use [9–12]. Thus, the demand for flashlamps as pump sources will be maintained in the future, and optimizing flashlamp-pumped lasers for better usability is a meaningful process. The latest issue with laser systems optimization is cost-effectively increasing the beam power and energy with system-scale miniaturization [13–18]. The gain module, commonly referred to as the pump chamber or laser head, is the most crucial laser system component. Thermal effects and energy storage characteristics for the gain module are essential to establish the structure and design parameters of the laser cavity and further to handle the power scaling and miniaturization. We suggest adopting these criteria for evaluating the performance of the gain module.

In this study, we proposed a procedure to evaluate the performance of the solid-state laser gain module, and it was verified by evaluating the flashlamp Nd:YAG laser head. The proposed procedure has extensive availability for various solid-state laser gain modules, including diode-pumping-type laser gain modules. We divided the gain module performance factors into two categories: thermal effect and energy storage characteristics.

The well-known variable that can best describe the performance of the gain module is the normalized heating parameter $\chi$, which is defined as the ratio of heat generated to energy stored in the pumped gain medium [19,20]. The thermal effects of the laser gain medium are the most significant impediments [21–25]. In high-power laser systems, especially, the issues of thermal stress on the laser gain medium causing depolarization loss, thermally induced birefringence, and thermal lens aberrations have been carefully addressed [26–30]. Moreover, since the laser cavity parameters such as output coupler, high reflectivity mirror, and cavity length depend on the heat dissipated into the gain medium, the precise analysis and measurement of thermal characteristics for the gain medium must be carried out to design a sophisticated cavity structure [19,31–33]. The energy stored in the pumped gain medium is related to system gain and laser output. The energy stored in the pumped gain medium is considered a design variable when simulating the laser pulse propagation and amplification in the laser amplifier [34]. If accurate information regarding the normalized heating parameter $\chi$ about the gain module can be provided to laser system developers, it would be helpful to prevent thermal problems in advance by predicting the heat, $Q$, generated in the laser crystal. Conversely, the energy storage properties, $\eta_G$, for the gain module can be estimated from $\chi$ and $Q$. So, one of the goals in our work was to quantify the heat, $Q$, generation inside the laser crystal to calculate the normalized heating parameter, $\chi$, as the performance indicator for the gain module.

Wavefront distortion caused by pumped laser gain medium has been a good reference for predicting thermal effects and output beam distortion [35–38]. We proposed a method to quantify the heat generation using not only the wavefront distortion caused by pumped laser crystal but also the thermal deformation theory of the gain medium and the ray transfer matrix method. The normalized heating parameter, $\chi$, and the thermal lens focal length have been calculated from the obtained heat quantity. The wavefront for the gain medium was obtained using a 4f imaging system and a wavefront sensor in the pump on/off circumstance [38]. From the optical path difference (OPD) between the non-pumped and pumped gain medium, one can identify the type of wavefront aberration that is inevitably generated due to the characteristics of the gain module and medium. It also provides the possibility to compensate for the distortion of the output beam. The suggested method has the advantage of measuring even during laser operation, since the wavefront imaging system can be located outside the laser system. Moreover, it is not sensitive to the quality of the probe beam because the dissipated heat is derived from the relative difference in wavefront in the pump on/off states. We measured the small-signal gain to find out the energy stored in the gain medium [20]. Furthermore, gain module efficiency, $\eta_G$, which can be a practical criterion, was defined as the ratio of the energy stored in the gain medium and the electrical input energy by integrating multiple theoretical efficiencies during the pump energy transfer process.

## 2. Theory and Method

### 2.1. Optical and Physical Distortion in Cylindrical Laser Rod

The thermal effect on a cylindrical laser rod $(r_0 \ll l)$ can be divided into optical and physical distortions, as shown in Figure 1 [19]:

1. $\Delta n_T, \Delta n_\varepsilon$: optical distortion by temperature and stress;
2. $R$: physical distortion by end surface curvature.

The optical distortion caused by changes in the temperature and stress leads to changes in the gain medium refractive index and can be expressed as

$$n(r) = n_0 + \Delta n(r)_T + \Delta n(r)_\varepsilon \tag{1}$$

where $n_0$ is the unperturbed refractive index of the gain medium, and $\Delta n(r)_T$ and $\Delta n(r)_\varepsilon$ are the changes in the refractive index owing to temperature and stress, respectively.

As shown in Equation (2), the effective refractive index of the thermal affected gain medium can be approximated as a parabolic distribution, the gradient index (GRIN),

$n_G(r)$, throughout the perpendicular plane of the rod. It represents a relative change for the refractive index on the optical axis of the rod, which is sufficient to apply to the methodology proposed in this study. This approximation was validated and utilized to model the thermal effects for the gain medium of the flashlamps pump scheme based on the assumption that the internal heat generation of the rod is uniform and rod length is much longer than the rod radius ($r_0 \ll l$):

$$n_G(r) = n_0 \left[ 1 - \frac{Q}{2K} \frac{1}{n_0} \left( \frac{1}{2} \frac{dn}{dT} + \alpha \, C_{r,\phi} \, n_0^3 \right) r^2 \right] \tag{2}$$

In Equation (2), $Q$ is the amount of heat dissipated into the pumped gain medium and is generally expressed in watts per unit volume. $K$, $dn/dT$, $\alpha$, and $C_{r,\phi}$ are the material properties, thermal conductivity, thermal coefficient of refractive index, thermal expansion coefficient, and photoelastic coefficients, respectively, as shown in Table 1. The photoelastic coefficients, $C_{r,\phi}$, of Nd:YAG are divided into radial ($r$) and tangential ($\phi$) components, depending on the polarization of light. The difference between the values of the two components causes thermal birefringence and a bifocusing effect on the pumped gain medium.

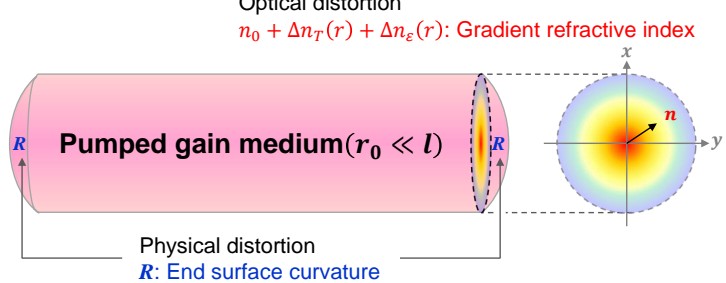

**Figure 1.** A pumped cylindrical rod can be modeled as a GRIN lens, where both end radii of curvature are $R$, by adding physical distortion to optical distortion($\Delta n(r)_T, \Delta n(r)_\varepsilon$) caused by the thermal effect.

**Table 1.** Material properties of Nd:YAG.

| | |
|---|---|
| Refractive index, $n_0$ (at 633 nm) | 1.83 |
| Thermal conductivity, $K$ (W/cm K) | 0.14 |
| Thermal coefficient of refractive index, $dn/dT$ (K$^{-1}$) | $7.3 \times 10^{-6}$ |
| Thermal expansion coefficient, $\alpha$ (K$^{-1}$) | $7.5 \times 10^{-6}$ |
| Radial photoelastic coefficient, $C_r$ | 0.017 |
| Tangential photoelastic coefficient, $C_\phi$ | $-0.0025$ |

The gradient index (GRIN), in the form of a quadratic function of the rod radius, can be expressed as:

$$n_G(r) = n_0 \left[ 1 - \frac{\left( \gamma_{r,\phi} \, r \right)^2}{2} \right] \tag{3}$$

The gradient constant, $\gamma_{r,\phi}$, can be expressed by Equation (4) using Equations (2) and (3).

$$\gamma_{r,\phi}(Q) = \sqrt{\frac{Q}{K} \frac{1}{n_0} \left( \frac{1}{2} \frac{dn}{dT} + \alpha \, C_{r,\phi} \, n_0^3 \right)} \tag{4}$$

The physical distortion appears as surface deformation at both ends on the pumped gain medium from the assumption that the thermal expansion only occurs at the end

section of the rod approximated by the $r_0$ scale, and it can be expressed as a single radius of curvature $R$:

$$R(Q) = \frac{2K}{\alpha Q r_0} \tag{5}$$

The thermal lens focal length for the cylindrical laser rod with the optical and physical distortion effect can be expressed as

$$f_{r,\phi}^{th} = \frac{K}{Ql} \left[ \frac{1}{2} \frac{dn}{dT} + \alpha\, C_{r,\phi}\, n_0^3 + \frac{\alpha r_0 (n_0 - 1)}{l} \right]^{-1} \tag{6}$$

$r_0$ and $l$ are the radius and length of the gain medium, respectively.

The following Sections 2.2 and 2.3 describe a method to derive the amount of heat dissipated into the gain medium experimentally by combining the wavefront difference in pumped and non-pumped states, the thermal deformation theory, and the ray transfer matrix method.

### 2.2. Gain Medium Surface Image Relay and Wavefront Reconstruction

A 4f image relay optical system, as shown in Figure 2a, was used to accurately image and measure the gain medium surface on the wavefront sensor plane. The 4f system can reconstruct the input field with high quality in the image plane and minimize the diffraction effect of the output field [39,40]. In addition, the optical system dimensions and magnification can be expressed by only using the focal lengths $(f_1, f_2)$ of the two positive lenses $(L_1, L_2)$, so it has the advantage of a relatively simple optical system design.

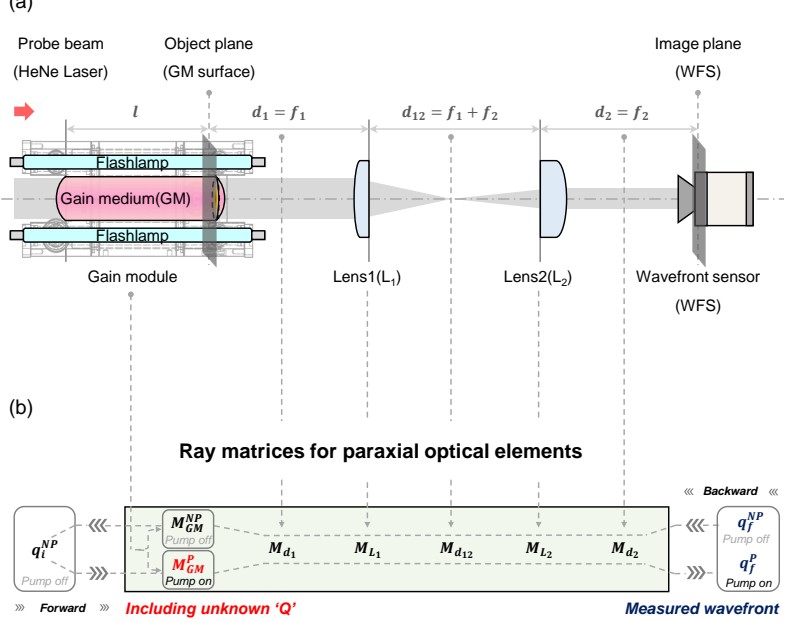

**Figure 2.** Conceptional scheme for the measurement of heat dissipated into the gain medium. (**a**) 4f optical imaging system for wavefront measurement of gain medium (GM) surface. GM right-side surface is object plane and is positioned at front focal plane of Lens1 $(L_1)$. Distance $(d_{12})$ between $L_1$ and $L_2$ is sum of focal lengths of two lenses. GM surface is imaged on wavefront sensor (WFS) plane (**b**) Ray transfer matrix and direction diagram for paraxial optical elements constituting optical system. $M_{d_1}$, $M_{d_{12}}$, and $M_{d_2}$ are ray matrices for free space between optics. $M_{L_1}$ and $M_{L_2}$ are ray matrices for image relay lenses. $NP$ and $P$ denote pump off and on states, respectively. GM left-side surface is indicated by lowercase letter $i$, and WFS position is defined as lowercase letter $f$. $M_{GM}^{NP}$ and $M_{GM}^{P}$ are corresponding to the ray matrices for GM as defined notations. $q_i^{NP}$, $q_f^{NP}$, and $q_f^{P}$ are complex beam parameters for each position and state.

The measured Zernike coefficient, $C_n^m$, for the gain medium through the wavefront sensor can be reconstructed as the wavefront, $\Phi$, in the polar coordinate system, $(\rho, \theta)$, by multiplying the Zernike polynomial, $Z_n^m$, for the analysis [41]:

$$\Phi(\rho, \theta) = \sum C_n^m Z_n^m(\rho, \theta) \tag{7}$$

*2.3. Ray Transfer Matrix to Obtain Heat Dissipated into Gain Medium*

Wavefront data measured using a Gaussian probe beam can be defined as a complex beam parameter, $q$, through a radius of curvature $R_C$, wavelength $\lambda$, refractive index $n$, and beam spot size $\omega$, which can be expressed as a ray vector, $\widetilde{q}$ [42]:

$$\widetilde{q} = \begin{pmatrix} q \\ 1 \end{pmatrix} = \begin{pmatrix} 1/R_C - i\lambda/\pi n \omega^2 \\ 1 \end{pmatrix} \tag{8}$$

As shown in Figure 2a, the direction from the wavefront sensor to the gain medium is defined as the backward direction. The ray transfer matrix for the backward-direction optical system is $M_B$ and can be expressed as Equation (9). $M_{T_b}$ in Equation (9) is the product of the ray matrices for the paraxial optical elements except for the gain medium. In the case of the 4f system, it can be simplified as Equation (10) using only the focal length of the image relay lens, $(f_1, f_2)$ [42]:

$$M_B = M_{GM}^{NP} M_{T_b} \tag{9}$$

$$\begin{aligned} M_{T_b} &= M_{d_1} M_{L_1} M_{d_{12}} M_{L_2} M_{d_2} \\ &= \begin{pmatrix} -f_1/f_2 & 0 \\ 0 & -f_2/f_1 \end{pmatrix} \end{aligned} \tag{10}$$

In Equation (9), $M_{GM}^{NP}$ is the ray transfer matrix for the non-pumped gain medium and can be considered as a free space consisting of the refractive index, $n_0$, for the center of the gain medium material. By multiplying the backward direction ray transfer matrix, $M_B$, and the ray vector, $\widetilde{q}_f^{NP}$, for the measured wavefront with normalization constant, $k$, the ray vector, $\widetilde{q}_i^{NP}$, for the non-pumped gain medium surface can be calculated:

$$\widetilde{q}_i^{NP} = k M_B \times \widetilde{q}_f^{NP} \tag{11}$$

The opposite way for the backward direction, i.e., the direction from the gain medium to the wavefront sensor, is defined as the forward direction, as shown in Figure 2b. The ray transfer matrix in the forward direction is $M_F$. Likewise, $M_{T_f}$ in Equation (12) is the product of the ray matrices for the 4f image relay, and it can be simplified as Equation (13):

$$M_F = M_{T_f} M_{GM}^P \tag{12}$$

$$\begin{aligned} M_{T_f} &= M_{d_2} M_{L_2} M_{d_{12}} M_{L_1} M_{d_1} \\ &= \begin{pmatrix} -f_2/f_1 & 0 \\ 0 & -f_1/f_2 \end{pmatrix} \end{aligned} \tag{13}$$

$$\begin{aligned} M_{GM}^P &= \begin{pmatrix} 1 & 0 \\ (1-n_0)/R & 1 \end{pmatrix} \\ &\times \begin{pmatrix} \cos \gamma_{r,\phi} l & (n_0 \gamma_{r,\phi})^{-1} \sin \gamma_{r,\phi} l \\ -(n_0 \gamma_{r,\phi}) \sin \gamma_{r,\phi} l & \cos \gamma_{r,\phi} l \end{pmatrix} \\ &\times \begin{pmatrix} 1 & 0 \\ (n_0 - 1)/-R & 1 \end{pmatrix} \end{aligned} \tag{14}$$

Unlike in the backward direction, $M_F$ in Equation (12) includes the ray transfer matrix, $M_{GM}^P$, for the pumped gain medium. As mentioned in Section 2.1, the pumped gain medium can be matrixed by applying the GRIN lens and the end surface curvature, as shown in Equation (14). The ray vector, $\tilde{q}_f^P$, can be calculated using the product of the forward ray transfer matrix, $M_F$, and the ray vector, $\tilde{q}_i^{NP}$, for the gain medium surface.

$$\tilde{q}_f^P = kM_F(\gamma_{r,\phi}, R) \times \tilde{q}_i^{NP} \tag{15}$$

The ray vector, $\tilde{q}_f^P$, can also be defined using the measured wavefront in the gain medium pumping state. Both the gradient constant, $\gamma_{r,\phi}$, and radius of curvature, $R$, are functions of the heat, $Q$, dissipated into the pumped gain medium, as shown in Equations (4), (5) and (14). Thus, only $Q$ is unknown in Equation (15). We used the least squares method to obtain heat, $Q$, and the details are described in Section 4.

The focal length of the thermal lens can be calculated by substituting the obtained heat, $Q$, into Equation (6). The radius of curvature, $R_C$, of the wavefront can be defined by the two axes, $(r, \phi)$, for the gain medium cross section [43]:

$$R_{C(r,\phi)} = \frac{d_{r,\phi}^2}{8P_{V(r,\phi)}} + \frac{P_{V(r,\phi)}}{2} \tag{16}$$

In Equation (16), $d$ and $P_V$ are the beam diameter and peak-to-valley value on the radial and tangential axes for the measured wavefront. Thus, the focal lengths of the thermal lens for the radial and tangential axes can be obtained.

### 2.4. Small-Signal Gain Measurement and Gain Module Efficiency

In the theory of laser pulse amplification, the small-signal gain, $G$, is defined as $\exp(n\sigma l)$ [19]. $n$ is the number of atoms per unit volume transferred from the ground level to the upper laser level in the gain medium during the pumping process, and $\sigma$ and $l$ are the emission cross section and length of the gain medium, respectively. $G(= E_{out}/E_{in})$ can be experimentally obtained by measuring the single-pass gain for a low input with the pumped gain medium, based on another definition of small-signal gain. By substituting the measured value, $G$, in the exponential formula of the small-signal gain, one can find the number of atoms, $n(= \ln G/\sigma l)$, in the upper state. The number of atoms, $n$, in the upper laser level can be expressed in terms of the energy stored, $E_{sto}$, in the gain medium by applying the photon energy formula, $(nh\nu)$, and gain medium volume.

The overall efficiency of a laser system is generally evaluated by integrating various efficiencies at each step from the electrical input to the laser output. This study focused on the energy flow from the pump source radiation to the upper laser level, and this process can be expressed as the product of five commonly known efficiency factors [19]:

$$E_{sto} = \eta_p \eta_t \eta_a \eta_Q \eta_S E_e \equiv \eta_G E_e \tag{17}$$

$E_e$ is the electrical energy supplied into the pump source. The pump source efficiency, $\eta_p$, is defined as the fraction at which the electrical input energy of the power supply is transferred as optical radiation corresponding to the gain medium absorption region. $\eta_t$ is the radiation transfer efficiency, which refers to the transmission efficiency of the radiation emitted from the pump source to the laser gain medium. $\eta_a$ is the absorption efficiency, defined as the ratio between the optical radiation entered to the gain medium and the amount absorbed. $\eta_Q$ is the quantum efficiency, which is defined as the ratio between the number of pump photons and photons contributing to laser emission. $\eta_S$ is the Stokes factor or quantum defect efficiency, which is the ratio of the pump transition wavelength to the laser emission wavelength of the gain medium. Combining the two factors $\eta_Q$ and $\eta_S$, it is called upper state efficiency, which is the ratio between the energy absorbed into the pump band and the laser transition. In this study, for a more practical perspective,

we define the gain module efficiency, $\eta_G$, as the ratio of the electrical energy input to the pump source and the energy stored in the gain medium using only the beginning and the end of the process. However, though it is challenging, measuring and distinguishing each theoretical efficiency can also be meaningful.

### 3. Experiment Setup

Figure 3a shows the experimental setup for thermal effect measurement on the gain module, the product of EKSPLA (2MA12-58) that pumps the gain medium through two flashlamps (VQX R, Verre and Quartz Flashlamps) in the vertical direction. The flashlamp driver was EKSPLA's PS5050 (PFN spec.: capacitance 60 µF, inductance 100 µH, and pulse duration FWHM 163 µs). The Nd:YAG (Laser Materials Corp.) was a [111]-cut rod with a diameter of 12.2 mm, length of 85 mm, and doping concentration of 0.8%. The temperature of the gain medium was maintained at 23 °C using a water chiller unit (P315, TERMOTEK). A collimated He-Ne laser (HRR120, Thorlabs) and a Gaussian beam of TEM$_{00}$ mode (>99%) was used as the probe beam. The refractive index change by thermal stress in Nd:YAG differs on the radial and tangential axes according to the polarization of the light, as mentioned in Section 2.1. The probe beam was polarized parallel to the horizontal axis for the incident plane of the rod. For the linearly polarized probe beam, the radial and tangential components, respectively, exist throughout the overall positions of the rod incident plane. However, for the incident light along the horizontal axis based on the defined coordinate axes in Figure 1, the radial components only needed to be applied in the y-axis, and the tangential components only needed to be applied to the x-axis. For measurement accuracy, a beam expander with a magnification of 16.6 (f2/f1 = 500 mm/30 mm) only allowed more than 80% of the Gaussian beam intensity to pass through the gain medium. A wavefront sensor (WFS150-5C, Thorlabs; aperture 5.95 mm × 4.76 mm; pixels resolution 1280 × 1024; exposure 76 µs–65 ms) was used for measuring wavefront aberration for thermally affected gain medium. Considering the diameter of the Nd:YAG and the sensor aperture size, the magnification (f2/f1 = 100 mm/250 mm) of the image relay was 0.4. Thermal relaxation time ($\tau' = r_0^2/4k_d$; $r_0$: rod radius, 12.2 mm; $k_d$: thermal diffusivity of Nd:YAG @ 300 K, 0.046 cm$^2$s$^{-1}$) [19] refers to the time it takes for the central temperature of the thermally affected rod to decay to 1/e. Applying our experimental conditions, this was about 2.02 s. In our experiment, the flashlamps emitted pump radiation for about 400 µs at a 10 Hz repetition rate, and the wavefront sensor exposure time was 37 ms. The sensor exposure time was only 1.83% compared to the thermal relaxation time of the gain medium. Thus, time-dependent measurement and analysis in the thermal relaxation time scale of the rod are possible, although a pulsed probe laser and shorter exposure times can be required to measure the thermal distortions in the gain medium in the more minute view during the pump time level. However, in this study, because the main research objective was to establish the methodology for the performance evaluation of the gain module, a measurement for the time dependency was not performed. The dissipated heat, normalized heating parameter $\chi$, and thermal lens were measured by adjusting the flashlamp driver voltage from 1.0 kV to 1.6 kV in 100 V units at a pulse repetition rate of 10 Hz.

Figure 3b shows the experimental setup for the small-signal gain measurement on the same gain module. A Quanta-Ray (LAB-150-10H) product (Spectra-Physics, Inc., Milpitas, CA, USA) was used as the seed laser. The flashlamp driver voltage was adjusted from 1.0 kV to 1.6 kV in 200 V units. The output energy for the low-energy seed beam of 6 µJ at 1064 nm was measured using an energy sensor (PE50-DIF-ER-C, Ophir).

(a)

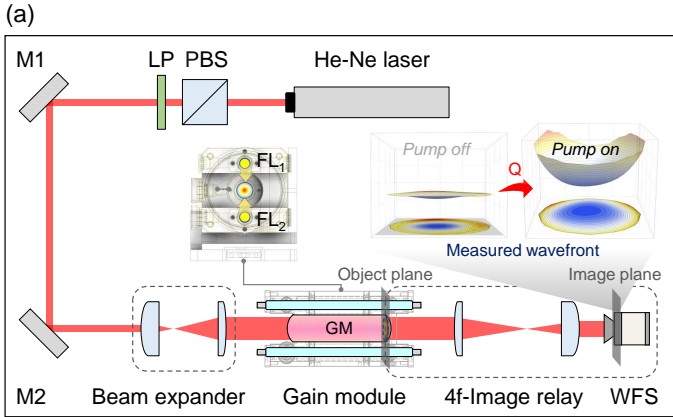

(b)

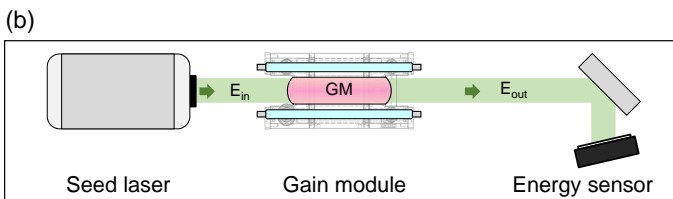

**Figure 3.** Experimental schematic for measuring (**a**) heat dissipated into the pumped gain medium. M1, M2: mirrors, LP: linear polarizer, PBS: polarizing beamsplitter, WFS: wavefront sensor, FL: flashlamp, GM: gain medium and (**b**) small-signal gain.

## 4. Results and Discussion

Figure 4a shows the Zernike coefficient measured by the wavefront sensor for the gain medium surface. The measured coefficients indicated in the blue series are data in the pump-off state and show the minor aberrations derived from the probe beam. Since the same probe beam was used even in the pumping state, measured coefficients indicated in the red series also included these aberrations. Thus, the distortion caused by the thermal effects can be identified from the difference in the measured values between pump on and off states, as shown in Figure 4b. The predictable piston and defocus aberrations owing to the gain medium thermal lens are remarkable, and as the pump input increases, the aberration also tends to increase. It was expressed only up to the second-order coefficient, since the coefficients for higher-order aberrations above the third-order term are close to zero. The Zernike coefficients for Tilt Y, Tilt X, Astigmatism 45°, and Astigmatism 0° were measured to be larger than zero but were also negligible quantities.

Figure 5a shows the reconstructed wavefront based on the measured Zernike coefficient at the gain medium surface without pumping. The wavefront for the perfectly collimated beam is flat, but the He-Ne laser used in the experiment has a divergence of approximately 0.92 mrad, so the collimation was not completely achieved even though the probe beam passed the beam expander and collimation systems. However, it is unnecessary to use the collimated probe beam because the heat dissipated into the gain medium is calculated from the relative amount of distortion in the pumped state compared to the wavefront in the absence of pumping. Figure 5b shows the wavefront for a 1.6 kV flashlamp input. Compared with Figure 5a, the wavefront is distorted by the thermal effect in the pumped gain medium. The wavefront of the pumped gain medium has a negative curvature; that is, the gain medium behaves as a positive thermal lens. Figure 5c,d are the extracted wavefronts on the vertical and horizontal axes for the gain medium cross section, from which the wavefront radius of curvature on the x and y axes can be obtained.

We used the least squares method to find the heat dissipated into the gain medium in Equation (15). As shown in Figure 6, one can find the heat, *Q*, value that minimizes the subtraction of the absolute square of right side, including the unknown *Q* in Equation (15) and the absolute square of $q_f{}^P$ for the measured wavefront in the pumped state.

Figure 7a shows heat, $P_h$, as a function of input voltage by multiplying the obtained heat, $Q$, with the gain medium volume. Because the electric energy ($CV^2/2$; $C = 60$ μF) delivered to the flashlamp is proportional to the square of the input voltage, the measured data were fitted to a quadratic function of the input voltage. The measurement uncertainty [44] of the heat, $P_h$, was calculated for five repeated measurements under the same input voltage conditions with $\pm0.1\%$ voltage stability and $\lambda/15$ RMS of wavefront accuracy. The maximum value was 0.67%.

Figure 7b shows the vertical and horizontal thermal lens focal length as a function of input voltage. The thermal lens exhibits astigmatism along the vertical and horizontal axes. It is a mixture of stress-induced birefringence and the nonuniform pumping scheme in which two flashlamps in the vertical direction pump the gain medium. In this study, it was not determined which of the two factors was dominant. Nevertheless, a research case shows that a suitable angle of polarization can minimize depolarization loss for the [111]-cut Nd:YAG crystal even in nonuniform pumping conditions [45]. Thus, in the follow-up research, measuring polarization dependency for the thermal lens can be meaningful to find a suitable angle that minimizes Nd:YAG thermal lens astigmatism.

Figure 8a shows the measured small-signal gain, $G$, as a function of the input voltage. Figure 8b, which concisely represents the conclusion of this study, shows the gain module performance as a normalized heating parameter, $\chi$, and gain module efficiency, $\eta_G$, describing the thermal effects and the energy storage characteristics. The gain module used in our experiment has a maximum efficiency of 3.02% when the input voltage is 1.2 kV. In contrast to the gain module efficiency, $\eta_G$, the normalized heating parameter, $\chi$, shows a minimum value of 1.85 at an input voltage of 1.2 kV. The experimental data are reliable because a high gain module efficiency indicates good energy storage characteristics with a lower thermal effect.

(a)

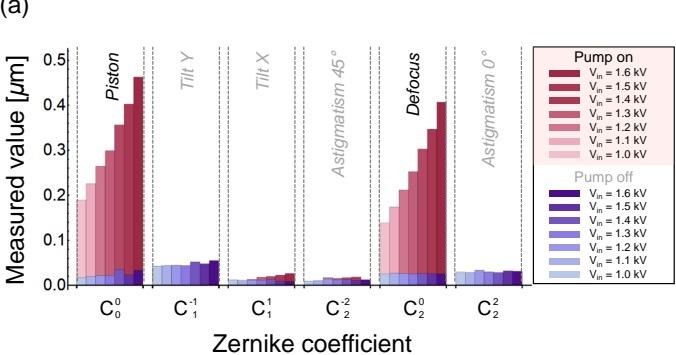

(b)

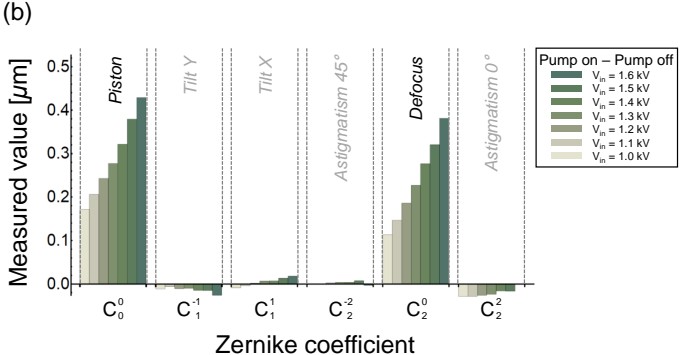

**Figure 4.** Measured Zernike coefficients, $C_n^m$, for the gain medium surface. The darker the tone, the higher the input voltage ($V_{in} = 1.0$ kV to 1.6 kV). (**a**) Blue series denotes pump-off state, and red series denotes pump-on state. (**b**) Difference of measured data in pump on and off states.

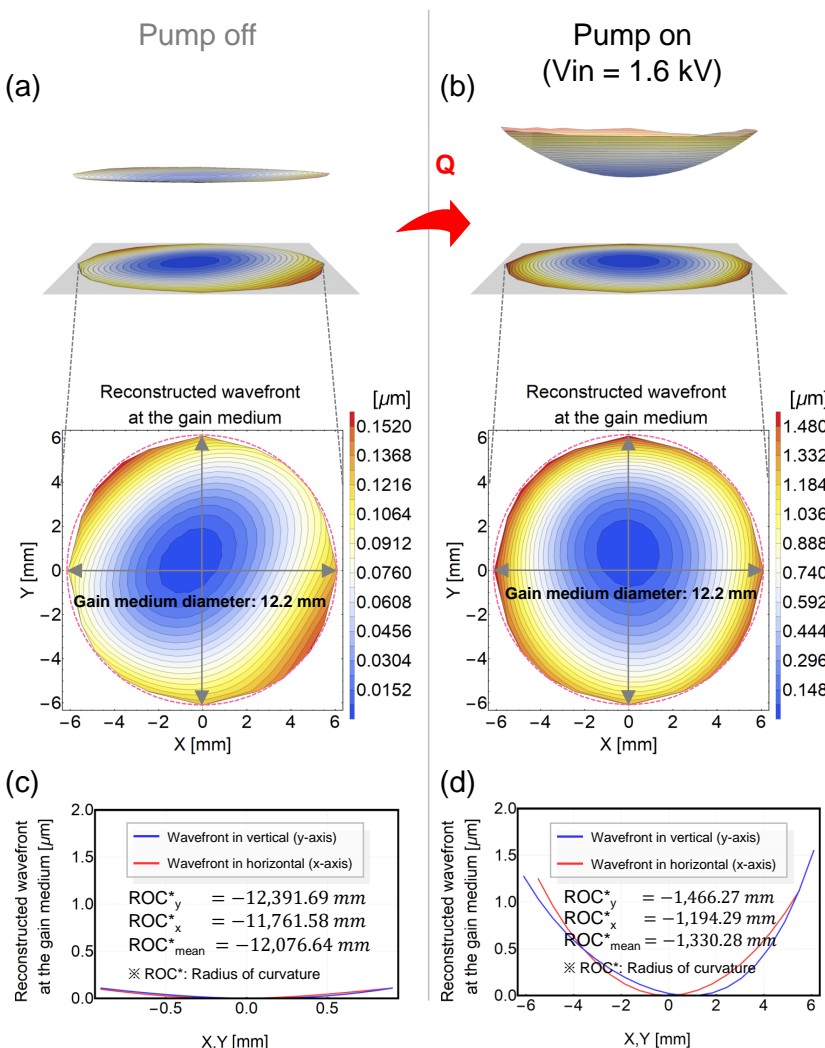

**Figure 5.** Example of reconstructed wavefront and extracted data for x ($\phi = 0$) and y ($\phi = \pi/2$) axes of gain medium cross section. (**a**,**c**) Pump off state. (**b**,**d**) Pump on state ($V_{in} = 1.6$ kV).

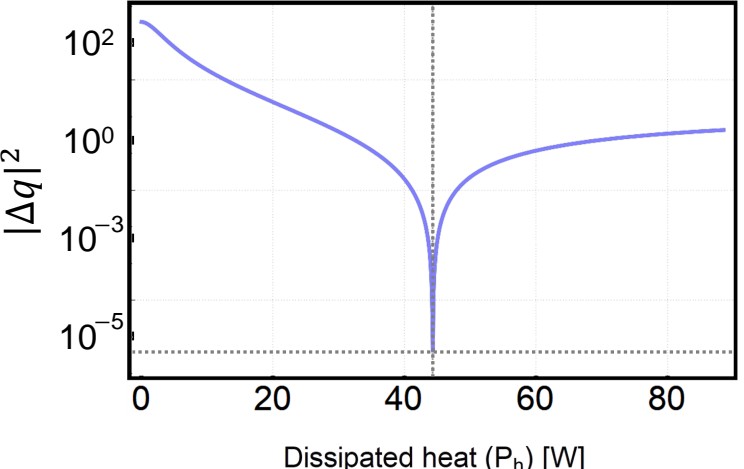

**Figure 6.** Example of determining heat, $Q$, dissipated into the gain medium, based on the least squares method, when flashlamp driver voltage is 1.6 kV. Absolute square of $|\Delta q|^2$ on vertical axis denotes the value obtained by subtracting $|(M_{F_{11}} q_i^{NP} + M_{F_{12}})/(M_{F_{21}} q_i^{NP} + M_{F_{22}})|^2$ from $|q_f^P|^2$ in Equation (15). $P_h$ [W] on the horizontal axis is heat, $Q$, multiplied by gain medium volume.

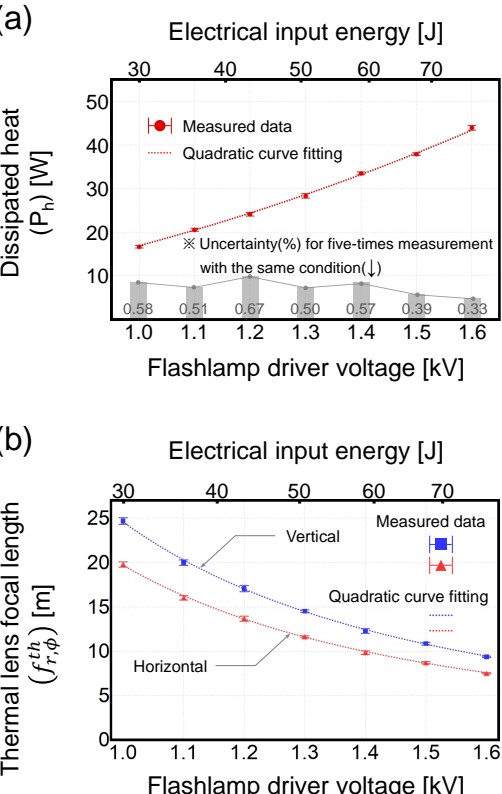

**Figure 7.** Result for measurement of gain module thermal effect. (**a**) Measured (red dot) and fitted (dashed red) data for heat dissipated into pumped gain medium (1.0 kV to 1.6 kV), measurement uncertainty with five repetitions (gray bar). (**b**) Measured (blue square, red triangle) and fitted data (dashed blue, dashed red) for thermal lens focal length (vertical, horizontal).

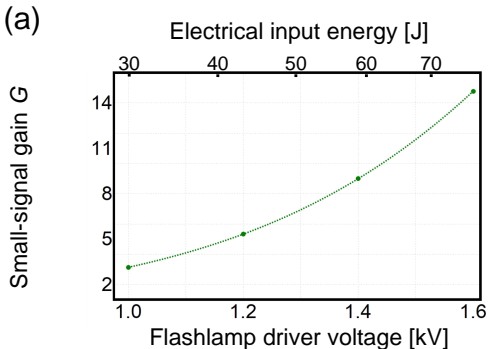

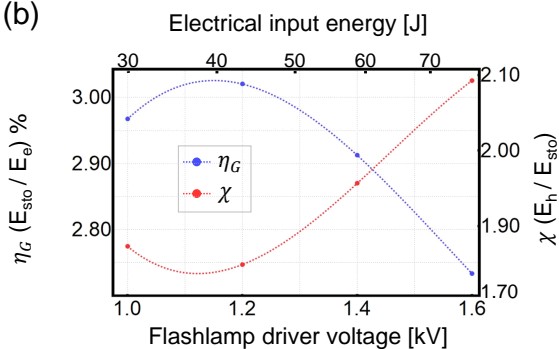

**Figure 8.** Result for measurement of (**a**) small-signal gain $G$ and (**b**) gain module efficiency $\eta_G$ (blue) and normalized heating parameter $\chi$ (red).

## 5. Conclusions and Outlook

We suggested the performance evaluation procedure for the solid-state laser gain module, which is the key component in laser systems. The measured data were sufficient to evaluate the performance of the gain module suggested as thermal effects and energy-storage features, with a simple measurement scheme in terms of practical aspects. We verified the proposed procedure by evaluating the flashlamp Nd:YAG gain module. We developed a method quantifying the heat dissipated into the gain medium to estimate the normalized heating parameter, $\chi$, using the measured wavefront in the pump on and off states, the gain medium thermal deformation theory, and the ray transfer matrix method. All of the measured data were expressed as functions of the input voltage of the pump. This datum type is beneficial to laser designers and technicians because the power supply voltage, directly correlated to the laser output, is mainly manipulated when setting up laser systems. The suggested method is straightforward because it only needs to measure the wavefront on the gain medium. One can identify potential aberrations that may affect the output beam from the measured wavefront distortion and find suitable alternatives. The image relay optical system for measuring the wavefront can be located outside the laser resonator, and it allows for the measurement of the thermal effect even when the laser is operating. It is also independent of the probe beam quality because the heat generation in the pumped gain medium can be obtained by analyzing the relative difference of the wavefront according to the presence or absence of the pump. In addition, the thermal lens focal length was calculated using the obtained heat. Thermal lens data showed astigmatism between the vertical and horizontal axes. The probe beam was p-polarized in this experiment, but if the thermal lens astigmatism can be changed according to the laser beam polarization, then the polarization direction with minimum astigmatism can be determined, and the thermal birefringence can be reduced, which is the next step of our research. The stored energy was obtained by measuring the small-signal gain according to the pump input voltage, and the energy storage of the gain module was evaluated by defining the gain module efficiency, $\eta_G$. We expect that securing the performance data for the gain module according to the procedure suggested in this study will be expanded into a prospective laser system design technique to improve beam power and quality by grasping the thermal effect and energy storage characteristics.

**Author Contributions:** Conceptualization, D.P. and J.J.; methodology, D.P. and J.J.; software, D.P. and J.J.; validation, D.P., S.C. and J.J.; formal analysis, D.P. and S.C.; investigation, D.P. and S.C.; resources, D.P., J.J. and S.C.; data curation, D.P., S.H. and S.L.; writing—original draft preparation, D.P. and S.C.; writing—review and editing, D.P., S.C. and T.J.Y.; visualization, D.P.; supervision, S.C. and T.J.Y.; project administration, T.J.Y.; funding acquisition, S.C. and T.J.Y. All authors have read and agreed to the published version of the manuscript.

**Funding:** This research was supported by Korea Institute for Advancement of Technology (KIAT) grant funded by the Korean Government (MOTIE) (P0008763, the Competency Development Program for Industry Specialist) and Basic Science Research Program through the National Research Foundation of Korea (NRF) funded by the Ministry of Education (2021R1I1A3051341).

**Institutional Review Board Statement:** Not applicable.

**Informed Consent Statement:** Not applicable.

**Data Availability Statement:** Not applicable.

**Conflicts of Interest:** The authors declare no conflict of interest.

## Abbreviations

The following abbreviations are used in this manuscript:

| | |
|---|---|
| OPD | Optical path difference |
| GRIN | Gradient index |
| GM | Gain medium |
| M | Mirror |
| LP | Linear polarizer |
| PBS | Polarizing beamsplitter |
| WFS | Wavefront sensor |
| FL | Flashlamp |

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
