# Peer review of "Performance Evaluation of Solid-State Laser Gain Module by Measurement of Thermal Effect and Energy Storage"

_photonics, doi:10.3390/photonics8100418_

Round 1
Reviewer 1 Report
The authors present a comprehensive study on the thermally induced distortions in a flash lamp pumped laser rod. Though the study describes certain gain module, the described procedure would be applicable as well for other gain modules and laser materials. The study is overall scientifically sound. Though the study does not introduce new findings, it would still be interesting for the community, as a basis for similar measurements.
Furthermore, I have the following remarks regarding the manuscript:
- The study relys on strongly simplified models. Though these should work for the given geometry, it would be good to have some comparisson with a more detailed model, e.g. a FEM-model. Especially one point is in my oppinion not absolutely correct: It is assumed that the thermal expansion is directly generating a radius on the rod'd surfaces. On the other hands the same expansion should lead to stress, again causing a thermal lens, but hindering the rods expansion. I think both effects will have an influence on each other.
- Please give an overview on the used parameters, like emission cross section, expansion coefficient, heat conductivity etc.. Such parameters often vary significantly in dependence on the literature used.
- Given the above mentioned points in my oppinion the retrieved heat deposition is quite error prone, though the dependencies would be correct. Nevertherless this should be discussed in more detail in the manuscript.
Author Response
First, we would like to appreciate the peer review of our manuscript entitled "Performance evaluation of solid-state laser gain module by measurement of thermal effect and energy storage"
After receiving and carefully reading the review reports, we considered any issues brought up by the reviewer and made necessary revisions to improve the manuscript. We respond/address all concerns and comments in a point-to-point manner. Specific answers to the reviewer's comments are attached. Please see the attachment.

Reviewer 2 Report
The paper describes the measurement of pump-induced wavefront change and single-pass gain in a flashlamp-pumped Nd:YAG laser rod and together with thermal deformation theory propose the evaluation of some performance parameters such as thermal lensing, heat dissipation, and energy storage. The authors suggest their method is useful for evaluation of solid-state laser performance and might be extendable to other solid-state materials and diode-pumping.
Whilst this paper contains some high-quality wavefront measurements and makes some good attempts to link this to theory, there are some issues with the analysis and many important parameters used in the fitting are missing from the paper. This missing data and some issues with the analysis make the paper less useful to the reader to apply to other system that the authors claim. Overall, I feel the paper is interesting and has some merit for publication but only after suitable addressing of the many issues detailed below.
The following points are mainly in order of appearance in paper, rather than importance:
Some factual errors to be corrected or needing clarification by authors:
- The parameter n0 is described as central refractive index in equation 1, but strictly should be unperturbed refractive index rather than n(0) as temperature and stress changes in refractive index (in eqn 1) are not zero, and often maximal, on axis. The authors could distinguish this term from the n0 used in equation 2 and 3 which is n(0).
- The formula used in equation 5 for surface radius of curvature R(Q) must be wrong as it is dimensionally incorrect, I believe. Equation 5 also suggests the surface bulge depends only on Q but has no relation to ratio of rod length and radius, but the integration of the surface bulge into the thermal lens focal length in equation 6 uses a different expression for R that does consider rod length and radius. This inconsistency should be sorted out by authors.
- In Section 3, the authors wrongly claim that the use of horizontal polarized probe sees only radial component of photoelastic effect. This is not true – it would only be correct for radially polarized light – linear polarized light “sees” different components of the radial and tangential photoelastic at different positions in the rod and is the basis for why you get depolarisation of light in Nd:YAG. The authors need to correct the false correlation they make of horizontal and vertical with radial and tangential, respectively.
- Related to comment 3, on page 10, authors propose the astigmatism in thermal lens “may be primarily affected by stress-induced birefringence” – this is clearly a wrong assertion based on misunderstanding as noted in comment 3. And their later comment to extend to probe with different polarization angles in future work should be reconsidered.
Missing Information in Paper that needs to be included:
- The authors take experimental data and compare to theory to extract some key information e.g. heat dissipation Ph(W), however, many material parameters of Nd:YAG are needed to derive these value e.g. Ph, χ. The authors should list in text or table all the material parameters they assume for in their analysis, including: thermal conductivity; dn/dT; thermal expansion coefficient; photoelastic coefficients; refractive index n0.
- Also state what are the efficiency factors assumed in Eqn 15? Most importantly, what is electrical to optical efficiency assumed and why do authors in eqn 15 take optical pump radiation energy rather than electrical for their analysis? They say it is approximately half electrical input, but realistically authors only know electrical energy. Also, all the efficiency factors come as a product so there is no way to untangle them.
- In experimental system state: WFS sensor size; lens f1 and f2 used in 4f image relay system
- A key omission is the relationship between flashlamp voltage and pump energy. The capacitor value should be stated as a minimum or the key final results in Fig.7a-d are ok in trend with voltage but are not quantifiable in absolute terms. I would recommend the authors either rescale x-axis in terms of pump energy or keep voltage axis and have an additional x-axis on top of figure graphs with pump energy.
Some other important considerations:
- The authors use a 4f imaging system in experiment which gives a scaled mapping of output wavefront to wavefront sensor. Why then do the authors add the complexity of 5 ray matrix matrices for this system when it can be described by a simple single matrix only dependent on f1 and f2 relay lenses? It would be helpful to reader to make this simplification.
- At end of page 9, The uncertainty 0.67% in the heat Ph derived measurement seems unreasonable, as it is likely dominated by experimental and Nd:YAG parameter uncertainties more so than shot to shot uncertainty.
- The final reports graphs Fig. 7a-d are too small, they were hard to read, especially with the information in the key under the curves on some of them. Given these are the key outputs of paper they should be made larger (maybe not use 2 graphs side-by-side).

Author Response

(The authors gave the same response as above.)

Reviewer 3 Report
The manuscript reports on an experimental work aimed at retrieving the "performance" (term used by the authors) of a flashlamp pumped rod Nd:YAG amplifier by measuring the wavefront aberrations induced on a "probe beam" passing through the crystal. In essence, the authors measure the wavefront distortions induced on a probe beam by the refraction index change occurring in the crystal due to the pumping. The authors then employ a simple model to retrieve the absorbed heat. First of all, it should be noted that the topic of the wavefront distortions induced by a pumped crystal on a beam is a well known and studied one, in particular in the context of ultrashort laser optics. Here the authors claim to be able to "evaluate the performance" of the module by just studying these distortions. First off, I have to say that the manuscript is well written and clear. Nevertheless, I have some major concerns about the real novelty and significance of the work, so that I would recommend its publication, unless a very extensive work is carried out on the manuscript.
1. It is not clear, in this context, what this "performance evaluation" actually consists in. Indeed, the authors are just measuring the transmitted wavefront of a probe pulse (by the way: most of the effects are well known, as I pointed out earlier). Even from a practical point of view, I don't understand how this work can be possiblt exploited.
2. My major concern is with the "time independent" character of the reported measurements. The authors use a CW beam as a probe, so that they carry out a time integrated measurement. In my opinion, it is not at all trivial how this measurement relates either to the "performance" of the gain module or to the wavefront distortions induced to the seed beam. I think that this issue (which includes, for instance, an evaluation of the thermal conductivity of the crystal, of the relaxation times, ...) must be discussed before considering the publication.
3. The model used to retrieve the heat Q assumes a uniform heat deposition and a parabolic profile of n in r. Although I can agree that this is, of course, a good approximation over a "small" region, I think the validity of this approximation should be discussed, as it may break even in the case of the case reported here.
Author Response

(The authors gave the same response as above.)

Round 2
Reviewer 2 Report
The revised manuscript meet my requested suggestions.
Reviewer 3 Report
I believe that the authors have seriously taken into account my concerns (in particular regarding the validity of their "time integrated" measurements) and addressed them in a satisfactory manner, at least making clear the limits of the technique. I can now recommend the publication.